# Comprehensive Analysis of the NF-YB Gene Family and Expression under Abiotic Stress and Hormone Treatment in *Larix kaempferi*

**DOI:** 10.3390/ijms24108910

**Published:** 2023-05-17

**Authors:** Lu Li, Xi Ren, Liying Shao, Xun Huang, Chunyan Zhang, Xuhui Wang, Jingli Yang, Chenghao Li

**Affiliations:** State Key Laboratory of Forest Genetics and Breeding, Northeast Forestry University, 26 Hexing Road, Harbin 150040, China; lilu96038@nefu.edu.cn (L.L.); rx2021121173@nefu.edu.cn (X.R.);

**Keywords:** NF-YB, *Larix kaempferi*, bioinformatics, transcription factors, abiotic stress

## Abstract

NF-YB, a subfamily of Nuclear Factor Y (NF-Y) transcription factor, play crucial role in many biological processes of plant growth and development and abiotic stress responses, and they can therefore be good candidate factors for breeding stress-resistant plants. However, the NF-YB proteins have not yet been explored in *Larix kaempferi*, a tree species with high economic and ecological values in northeast China and other regions, limiting the breeding of anti-stress *L. kaempferi.* In order to explore the roles of NF-YB transcription factors in *L. kaempferi*, we identified 20 *LkNF-YB* family genes from *L. kaempferi* full-length transcriptome data and carried out preliminary characterization of them through series of analyses on their phylogenetic relationships, conserved motif structure, subcellular localization prediction, GO annotation, promoter *cis*-acting elements as well as expression profiles under treatment of phytohormones (ABA, SA, MeJA) and abiotic stresses (salt and drought). The *LkNF-YB* genes were classified into three clades through phylogenetic analysis and belong to non-LEC1 type NF-YB transcription factors. They have 10 conserved motifs; all genes contain a common motif, and their promoters have various phytohormones and abiotic stress related *cis*-acting elements. Quantitative real time reverse transcription PCR (RT-qPCR) analysis showed that the sensitivity of the *LkNF-YB* genes to drought and salt stresses was higher in leaves than roots. The sensitivity of *LKNF-YB* genes to ABA, MeJA, SA stresses was much lower than that to abiotic stress. Among the *LkNF-YBs*, *LkNF-YB3* showed the strongest responses to drought and ABA treatments. Further protein interaction prediction analysis for *LkNF-YB3* revealed that *LkNF-YB3* interacts with various factors associated with stress responses and epigenetic regulation as well as NF-YA/NF-YC factors. Taken together, these results unveiled novel *L. kaempferi* NF-YB family genes and their characteristics, providing the basic knowledge for further in-depth studies on their roles in abiotic stress responses of *L. kaempferi*.

## 1. Introduction

Nuclear Factor Y (NF-Y) is a transcription factor superfamily, which plays crucial roles in plant growth, development and responses to environmental stresses. It is also called a family of the Heme Activated Protein (HAP), which can recognize and bind to CCAAT (a consensus sequence) and is therefore widely known as the CCAAT Binding Factor (CBF). The NF-Y family has high specificity and affinity, that can be further divided into three distinct subunits: NF-YA, NF-YB and NF-YC [1,2].

The NF-YB transcription factor, a subunit encoding the CCAAT Binding Factor (CBF), is a highly conserved trimer activator that binds strongly to CCAAT box promoter elements. NF-YB regulatory genes are ubiquitous among eukaryotes; however, there is only one gene encoding NF-YB in yeast and animals [3]. In contrast, plant genes encoding the NF-YB subunit are expanded, such as 34 NF-YB genes in wheat [4] and more than 10 in *Arabidopsis*, Z. jujuba and rice [5,6,7]. The NF-YB subunit consists of three domains, including amino-terminal A domain, a central conserved B domain and carboxyl-terminal C domain [8]. Of these, the B domain contains the amino acid residues necessary for interaction with NF-YA and NF-YC [8,9], and according to whether the B domain contains 16 amino acid residues (MIRHDIYNDRQTAEKT) [10,11] or not, the NF-YB subunits are classified into either LEAFY COTYLEDON1(LEC1) or non-LEC1 types. Some studies have suggested that the non-LEC1 NF-YB genes may evolve from duplication and functional divergence of LEC1 NF-YB genes [12,13,14].

In plants, NF-YBs are well-known transcription factors, which play multiple roles in regulating various aspects of growth and development, especially the reproduction, photosynthesis, phytohormone signal transduction and abiotic stress resistance [15,16,17,18]. Previous studies have provided a large body of evidence showing that NF-YB proteins are the key transcription factors for plant reproduction, particularly the flowering and embryogenesis and phytohormone signaling and abiotic stress responses. For example, *AtNF-YB*2 stimulates flowering by targeting key flowering genes, such as Flowering locus T (FT) and Suppressor of Constans 1 Overexpression (SOC1) [19]. Additionally, *AtNF-YB2* and *AtNF-YB3*, in collaboration with *AtNF-YC3*, *4* and *9*, regulate the photoperiod pathway, thereby affecting flowering time [20]. *AtNF-YB*9 has been reported to be an important regulator of zygotic embryogenesis, seed maturation and fatty acid synthesis in both mutant and wild-type *Arabidopsis* thaliana [21,22]. Overexpressing *AtNF-YB*9 (AtLEC1) in transgenic *Arabidopsis* promoted the formation of embryo-like structures in leaves through activating embryo-specific gene expression [23]. *PpNF-Ys* were specifically expressed in reproductive tissues, implying that they might participate in the development of floral organs and fruits [24]. NF-YB proteins also regulate photosynthesis in plants. Overexpression of *ZmNF-YB16* could improve photosynthesis and anti-oxidation ability of maize [25]. Moreover, plant NF-YB proteins are involved in phytohormone (i.e., ABA) signaling transduction and abiotic stress responses. In *Arabidopsis*, NF-YB3 forms a trimer with NF-YC2 and NF-YA4, which then binds to *AtbZIP28*, constituting a heterodimer that targets the promoters of ABA-response genes [26]. Other NF-YB proteins, such as *AtNF-YB*1, have been found to be involved in drought tolerance and abscisic acid signaling transduction [27]. *StNF-YB3.1* has been demonstrated to regulate ABA-dependent response against drought stress in potato [28,29]. Ectopic overexpression of poplar *PdNF-YB7* in *Arabidopsis* up-regulated downstream genes of the ABA-dependent pathway, thereby enhancing tolerance against drought [30]. Soybeans *GmNFYA3* also increased drought tolerance in an ABA-dependent manner [31]. Furthermore, *SiNF-YB8* has been shown to improve drought and osmotic tolerance in tobacco [32]. Overexpression of *PwNF-YB3* in *Arabidopsis* can significantly improve tolerance to drought and osmotic stress [33]. These findings have confirmed that plant NF-YB proteins are important transcription factors that influence not only development but also resistance to abiotic stresses.

*L. kaempferi*, a conifer species, mainly populate northern China, eastern Europe and western North America. *L. kaempferi* is commonly known as larch trees, with high value for wood production and ecological afforestation, which has several advantages over other larch trees, such as juvenile growth, wood with higher quality and greater resistance to environmental changes. However, recently recurring abnormal climate results in abiotic stresses, such as drought, severely challenging the afforestation practices of *L. kaempferi*. It is therefore of great importance to breed novel *L. kaempferi* varieties with improved abiotic stress resistance. The NF-YB gene family is a good candidate gene family, which can be utilized to breed stress-resistant *L. kaempferi* varieties. In this study, we identified NF-YB family genes in *L. kaempferi* and carried out comprehensive analyses for the first time. We identified a total of 20 *LkNF-YB* family genes and performed multiple analyses on their phylogeny, gene structure, conserved motif and promoter cis-elements, as well as GO ontology annotation. In addition, we selected six *LkNF-YB* genes of this family and studied their expression pattern in roots and leaves under various stress conditions, such as salt and drought, as well as phytohormone (ABA, SA and MeJA) treatment. The result indicated that *LkNF-YB* transcription factors play roles in regulating *L. kaempferi* responses to phytohormone and abiotic stresses. Our work provides a comprehensive overview of the *L. kaempferi* NF-YB gene family and lays the foundation for further research on the role of NF-YB factors as well as efforts to improve *L. kaempferi* stress resistance by regulating NF-YB factors.

## 2. Results

### 2.1. Identification and Sequence Analysis of NF-YB Genes in L. kaempferi

We identified *LkNF-YB* genes in *L. kaempferi* full-length transcriptome data by pfam analysis and HMMER. Through presorting and manual deletion of the redundant sequences, we identified 20 *LkNF-YB* genes. Then, we named these genes *LkNF-YB1* to *LkNF-YB20* and calculated the protein length of the 20 genes, which ranged from 151 amino acids to 316 amino acids. Molecular weights range from 16 kDa to 34 kDa, with isoelectric points ranging from 4.69 to 8.93. All 20 *LkNF-YB* proteins showed negative GRAVY values, which means that they are hydrophilic proteins. Subcellular localization prediction analysis shows that *LkNF-YB* proteins are localized to nucleus, chloroplast, mitochondria, extracellular and cytoplasm (Table 1).

### 2.2. Phylogenetic Relationship and Conserved Motifs of LkNF-YB Proteins

We constructed a phylogenetic tree using four species of NF-YB protein sequences from *L. kaempferi*, *Arabidopsis thaliana*, *Populus trichocarpa* and *Triticum aestivum*. An unrooted tree was constructed with the maximum likelihood (ML) method, and its evolutionary relationship was analyzed and visualized by MEGA 7.0. The 20 *LkNF-YBs* were divided into three clades (Figure 1). *LkNF-YB1*, *2*, *16*, *17*, *18*, *19*, *20* proteins were clustered in clade 1, suggesting that these *LkNF-YB* proteins were closely related. *LkNF-YB4*, *5*, *6*, *7*, *8*, *9*, *10*, *11*, *12*, *13* proteins were closely related in clade 2, and *LkNF-YB3*, *14*, *15* were classified in clade 3. We also performed MEME motif analysis and found that *LkNF-YB* contains 10 conserved motifs; we named them motif 1 to motif 10. Combined with the phylogenetic relationship of the *LkNF-YB* family (Figure 2a), we found that all *LkNF-YBs* contain motif 1 (Figure 2b).

### 2.3. Sequence Analysis of LkNF-YB Proteins

Multiple sequences alignment of the full-length amino acid sequences of *LkNF-YB* proteins indicated that *LkNF-YB* proteins have NF-YB family-specific B domain. There is clear evidence that they share a common motif sequence in the B domain: c-[VA]-[ST]-e-X-i-S-f-[LIVM]-t-[SGC]-e-a-[SCN]-[DE]-[KRQ]-C (Figure 3). It can be seen that all 20 *LkNF-YB* proteins were non-LEC1 type.

### 2.4. Analysis of Promoter Cis-Elements

We utilized online database PlantCARE to analyze hormone-related and stress-related *cis*-elements in the 2000 bp region upstream of the *LkNF-YB* genes to understand the possible regulatory mechanisms of these genes under hormone treatment and abiotic stresses. Ten different *cis*-elements were identified, such as the auxin-TGA-element, abscisic acid-ABRE, gibberellin-GARE-motif, P-box, salicylic acid-TCA-element, and MeJA-TGACG-motif responsive elements, and drought-MBS, defense-TC-rich, light-G-box, TCT-motif, low temperature-LTR, wound-WUN-motif responsive elements (Figure 4), suggesting the putative roles of *LkNF-YB* genes in phytohormone signaling and abiotic stress responses of *L. kaempferi*. It was found that all *LkNF-YB* genes share light response elements. Among *LkNF-YB* genes, the *LkNF-YB7* gene promoter has the most *cis*-elements.

### 2.5. Gene Ontology (GO) Annotation

The GO analysis of the *LkNF-YB* genes revealed the cellular environment in which they may perform molecular functions, as well as the biological processes involved (Figure 5). The *LkNF-YB* genes were mostly found in the categories of ‘cellular, metabolic and biological processes’, followed by response to stimulation, multicellular biological processes, participation in reproduction and reproductive processes. Only 5% of *LkNF-YB* genes are involved in growth. In terms of the ‘molecular function’ category, 18 *LkNF-YB* genes were involved in transcriptional regulatory activity. Cell composition prediction showed that 5% of the *LkNF-YB* genes belong to the category of ‘anatomical body of the cell’, whereas 90% of the *LkNF-YB* genes belong to the ‘protein complex’, two of which are part of the membrane.

### 2.6. Expression Pattern of LkNF-YB Genes under Salt, Drought, ABA, SA and MeJA Treatments

According to the preliminary treatment analysis of salt, drought, ABA, SA and MeJA, *LkNF-YB1*, *3*, *8*, *13*, *14*, *17* showed significant differential expression among 20 *LkNF-YB* genes. We then investigated the responses of these six genes under salt, drought, ABA, SA and MeJA treatment (Figure 6). First, we investigated the effect of abiotic stresses, such as salt and drought, on the *LkNF-YB* genes. We analyzed the expression levels of six genes in the leaves and roots of *L. kaempferi* after 3, 6, 12 and 24 h of 150 mM NaCl treatment. In leaves, all genes showed significant responses at all time points. Among them, remarkably strong responses were observed for *LkNF-YB3*, *17* under salt condition, as well as for *LkNF-YB1*, *8*, *13*, *17* under drought condition. The responses of *LkNF-YB1*, *8*, *13*, *17* against drought were induced at the highest level after drought treatment for 6 h. Only the *LkNF-YB17* was highly induced under salt condition as early as in drought. Except for *LkNF-YB3* and *13*, the expression levels of *LkNF-YB1*, *8*, *13* and *17* decreased with the extension of time. In roots, *LkNF-YB3*, *13*, *14* showed the strongest responses under salt and drought conditions. Among them, *LkNF-YB3* was strongly induced at 6, 12, 24 h under both salt and drought conditions. The responses of all six *LkNF-YB* genes except *LkNF-YB1*, *3* against salt and drought stress were not strong at the early stage, such as 3 h, and were then strengthened as the salt and drought stress lasted longer, suggesting that it takes time of over 3 h for stress signaling to activate *LkNF-YB* genes. Additionally, it was obvious that *LkNF-YB* genes were more highly induced by drought than salt stress. The promoters of six genes have MBS homeopathic elements, which have previously been known to play a role in regulating response to drought. In leaves, the expression of the *LkNF-YB17* gene was up-regulated at all time points compared to control under drought condition. While *LkNF-YB1*, *8* and *14* were down-regulated at 3 h, *LkNF-YB8* and *13* were up-regulated at 6, 12 and 24 h. The expressions of *LkNF-YB1*, *3*, *8*, *14* and *17* all increased first to the peak at 6 h and decreased later. In roots, relatively strong responses to drought were found in *LkNF-YB1*, *3*, *13* and *14*, among which *LkNF-YB3* and *14* showed the strongest responses at 6, 12 and 24 h with the increasing expression levels. *LkNF-YB1*, *3*, *8* are the genes that responded at 3 h, earlier than others. Overall, six *LkNF-YB* genes all showed significant responses to abiotic stresses, such as salt and drought, and their responses were all different according to the organ type (leaves or roots) and time points (3, 6, 12 or 24 h). Moreover, we confirmed that treatments with hormones, such as ABA, SA and MeJA, also differentially induced organ-type- and time-specific responses of *LkNF-YB* genes. In leaves, *LkNF-YB* genes were expressed at the highest levels under the ABA treatment, differently to the SA and MeJA treatments. *LkNF-YB1*, *3*, *17* showed high up-regulation upon hormone treatments, which was not the case for the remaining genes. The responses of the *LkNF-YB* genes to ABA were earlier and stronger than those to other hormones. In roots, only the *LkNF-YB3* showed strong responses to hormone treatment, and ABA induced its expression at all the time points. Taken together, *LkNF-YB* genes are mostly found to be associated with drought and ABA, implying their roles in regulating drought responses in an ABA-dependent manner. (The relevant data are attached in Appendix A of Appendix A).

### 2.7. Protein Interaction Network of LkNF-YB3

We further identified 28 putative interacting proteins for *LkNF-YB3* based on the criteria of ‘coexpression degree > 0.2’ (Figure 7). They include phytoene synthase, PsbQ subunit of photosystem II, phosphoribulokinase, NF-YC2, NF-YC10, HSP proteins (Heat shock protein 70-2, Heat shock protein 70 and ER-localized member of the HSP70), histone related proteins (Histone superfamily protein, histone H2A protein, histone H2A protein, histone H2A protein, histone H2A protein, histone H2A protein and Histone superfamily protein), 3-hydroxy-3-methylglutaryl coenzyme A reductase, cytochrome b6f complex subunit (petM), alpha/beta-Hydrolases superfamily protein, farnesyl diphosphate synthase, Pyruvate dehydrogenase kinase, Dwarf5, calcium homeostasis regulator, Photosystem II 5 kD protein, Rubredoxin-like superfamily protein, Peptidyl-Prolyl Isomerase, TPR-like superfamily protein and FKBP-type peptidyl-prolyl *cis*-trans isomerase. These proteins are mainly associated with stress responses and epigenetic regulation, indicating that stress responses activated by *LkNF-YB3* could be achieved through epigenetic regulation, such as histone modification.

Moreover, we predicted inter-family interaction between the NF-Y family proteins (Figure 8). NF-YB3 interacts with various NF-YA and NF-YC family proteins, including NF-YA1, NF-YA2, NF-YA3, NF-YC1, NF-YC2, NF-YC3, NF-YC4, NF-YC9 and NF-YC10.

## 3. Discussion

In plants, the NF-YB gene family encodes transcription factors, which are involved in various important biological processes, such as reproduction and abiotic stress responses. There has so far been growing evidence for the roles of the NF-YB gene family in several plant species, including wheat, apple, Populus and *Arabidopsis* [34,35,36,37,38]. In a non-tree model plant, such as *Arabidopsis*, NF-YB subunits 2 and 3 were found to promote *Arabidopsis* reproduction in response to light [38]. In an important agricultural crop plant such as wheat, NF-YB4 was also proved to be involved in the regulation of reproduction, which could contribute to improvement in grain yield [35]. In addition, NF-YB family factors in wheat and tree plants, including apple and poplar, have been characterized as regulating the plant responses to various environmental cues, such as abiotic stresses or phytohormones [35,36,37]. In this paper, we, for the first time, identified the NF-YB gene family in *L. kaempferi,* which has high economic and ecological values in northeastern China. We carried out multiple analyses on their phylogenetic relationships, conserved motifs and promoter *cis*-elements, as well as GO ontology annotation. Then, we investigated whether abiotic stresses and phytohormones affect the expression of *LkNF-YB* gene family or not.

Gene family identification and phylogenetic analysis showed that the *L. kaempferi* NF-YB gene family includes 20 NF-YB genes, and they have a close evolutionary relationship with the *AtNF-YB, PtNF-YB* and *TaNF-YB* gene family (Figure 1). Motif analysis revealed that *LkNF-YBs* shared highly conserved amino acid sequences of total 10 motifs (Figure 2). *LkNF-YB* proteins all have c-[VA]-[ST]-e-X-i-S-f-[LIVM]-t-[SGC]-e-a-[SCN]-[DE]-[KRQ]-C, which is known as the NF-YB family-specific B domain (Figure 3). As this B domain does not contain 16 amino acid residues (MIRHDIYNDRQTAEKT), *LkNF-YB* proteins belongs to the non-LEC1 family. Since the *L. kaempferi* full-length transcriptome data we studied only included CDS, we could not analyze the intron–exon structure. Subcellular localization prediction analysis showed that *LkNF-YB* proteins are mostly found in the nucleus, indicating they are the transcription factors. Studies have shown that many plant TFs promoters have highly conserved *cis*-elements playing crucial roles in transcriptional regulation signaling pathways when plants are subjected to biological and abiotic stresses [39]. Therefore, *cis*-acting elements in the promoter region of *LkNF-YB* genes were analyzed, and it was found that every gene contained at least three elements related to hormonal or biological stress. In addition to the light response elements, drought response elements and ABA induction associated elements accounted for the largest proportion, indicating that NF-YB genes can be highly responsive to drought and ABA, which has been confirmed in other plants [40]. For example, drought and ABA have been demonstrated to induce GmNFYB17 expression, showing a consistent pattern of increased expression in soybean [41]. Another recent study showed that PdNF-YB21 positively regulated poplar drought stress tolerance through ABA [42]. The GO analysis of *LkNF-YB* genes revealed their role in a variety of biological processes, such as metabolic responses and stimulus regulation.

We also demonstrated the response of *LkNF-YB* family genes to phytohormones (ABA, SA and MeJA) and abiotic stresses (salt and drought). A total of 20 genes were identified in the study. RT-qPCR showed that most *LkNF-YB* genes were up-regulated under different stress treatments. The sensitivity of the *LkNF-YB* genes under drought stress was higher in roots than leaves, while the sensitivity of LkNF-YB genes under salt stress was the opposite, and the leaf sensitivity was higher under hormones, such as ABA, MeJA and SA stress. Interestingly, we found that the expression levels of some genes under salt and drought stress were consistent with those after ABA treatment. The expression patterns of *LkNF-YB 1* and *3* were similar after ABA and salt treatment, while the expression patterns of *LkNF-YB 14* and *LkNF-YB 17* were similar after ABA and drought treatment. Combined with the results of *cis*-element analysis, cross-regulation between the ABA signaling pathway and response to drought and salt stress can be inferred, but its correlation needs further study. The expression levels of *LkNF-YB* genes were significantly different under the three hormone treatments, and they were obviously up-regulated after the ABA treatment, while some genes were down-regulated under the other two hormone treatments, such as *LkNF-YB*1 and *LkNF-YB8* in roots.

Significant and strong responses under abiotic stress and hormone treatment were only found in six *LkNF-YB* genes, including *LkNF-YB1*, *3*, *8*, *13*, *14*, *17*. Therefore, the expression levels of six *LkNF-YB* genes in response to NaCl, PEG, ABA, SA and MeJA were analyzed. Consistent with previous studies and prediction from the promoter *cis*-acting element analysis results, six *LkNF-YB* genes all revealed significant responses against hormones and stress conditions. The strongest responses were found in *LkNF-YB3* under drought and ABA treatments.

We further predicted the protein interaction network for *LkNF-YB3* and found that *LkNF-YB3* interacts with multiple histone proteins, such as Histone superfamily protein, histone H2A protein, histone H2A protein, histone H2A protein, histone H2A protein, histone H2A protein and Histone superfamily protein.

So far, little research has been conducted on how NF-YBs affect epigenetic regulation in plants. However, there has been emerging evidence that NF-Y TFs mediate installing epigenetic marks that regulate the response to environmental or intrinsic signals in plants [43]. NF-Y proteins are mainly regarded as photoperiod-dependent flowering regulators conserved in eukaryotes [44,45,46,47,48]. The CONSTANS (CO)-mediated photoperiod pathway and GA signaling function in parallel promote flowering during long days (LDs), and these two pathways are integrated by NF-Y proteins. The NF-YB and NF-YC subunits are known to have conserved histone fold motifs that resemble H2B and H2A, respectively. These two are the core histone proteins of the nucleosome, and they are the fundamental unit of chromatin [49]. In yeast and animals, NF-Y proteins are known to modulate the chromatin structure and function through replacement of H2A–H2B and/or covalent modifications of histones, such as methylation and acetylation. In plants, the NF-Y complex binds to the SOC1 promoter to modulate trimethylated H3K27 levels in cooperation with a H3K27 demethylase REF6 [50]. In this work, the prediction results of *LkNF-YB3* showed that *LkNF-YB3* interacts with multiple histone proteins of Histone superfamily protein, histone H2A protein, histone H2A protein, histone H2A protein, histone H2A protein, histone H2A protein and Histone superfamily protein, providing consistent information on histone-related roles for NF-Y proteins.

## 4. Materials and Methods

### 4.1. Data Collection and Identification of LkNF-YBs

Thirteen NF-YB protein sequences (AT2G38880, AT5G47640, AT4G14540, AT1G09030, AT2G47810, AT5G47670, AT2G13570, AT2G37060, AT1G21970, AT3G5340, AT2G27470, AT5G08190, AT5G23090) were retrieved from the *A. thaliana* database (https://www.Arabidopsis.org/ (accessed on 25 January 2022)). The hidden markov models of the NF-YB proteins were obtained from the Pfam database (Pfam: Home page (xfam.org)(accessed on 1 February 2022)) [51]; the HMM file of NF-YB was downloaded (PF00808). The large genome size and complex genetic background of *L. kaempferi* [52] make the currently sequenced *L. kaempferi* whole genome sequence and its annotation information too complex to be used for gene family identification analysis. Thus, in HMMER 3.0 (http://www.hmmer.wustl.edu/ (accessed on 3 February 2022)), the *L. kaempferi* full-length transcriptome data were utilized as a base database to retrieve and obtain the reference sequence [53]. The obtained sequences were predicted using Batch SMART of TBtools (version number 1.098769) and compared with the amino acid sequences of *A. thaliana* NF-YB [54]. According to the conserved domain of plant NF-YB, a total of 20 NF-YB genes were obtained by manually deleting the redundant sequences. The information on *LkNF-YBs* sequences, such as amino acid number, molecular weight, isoelectric point and hydrophilic mean value, was obtained by using the ProtParam tool of the ExPASy program (https://web.expasy.org/protparam/ (accessed on 11 February 2022)) [55]. We predict the cellular location of *LkNF-YBs* using the online tool WOLF PSORT (https://wolfpsort.hgc.jp/ (accessed on 15 February 2022)) [56].

### 4.2. Analysis of Conserved Motifs of LkNF-YB Proteins

The conserved motifs of *LkNF-YBs* were predicted using the online databaseMEME (http://meme-suite.org/tools/meme (accessed on 10 March 2022)) with the default parameter and conserved motif number of motifs set to 10 [57]. TBtools (version number 1.098769) was then used to visualize the evolutionary tree and conserved motifs of *LkNF-YBs*.

### 4.3. Multiple Sequence Alignment and Phylogenetic Analysis of LkNF-YBs

The amino acid sequences of *LkNF-YBs* were compared using Clustal X 2.0, and the results were plotted with GeneDoc. The conserved sequences of the NF-YB proteins of *L. kaempferi*, *A. thaliana*, *P. trichocarpa* and *T. aestivum* were detected with the MEGA7.0 software to construct a phylogenetic tree (no. of bootstrap replications = 500) with the neighbor-joining (NJ) method [58]. Then, use the online database evolview beautification (https://evolgenius.info/evolview (accessed on 25 March 2022)).

### 4.4. Promoter Cis-Element Analysis

As *L. kaempferi* full-length transcriptome data hold incomplete promoter information for genes, the online database NCBI BLAST (BLAST: Basic Local Alignment Search Tool (Nih.gov)) was used to search for *LkNF-YB* gene promoters. Online database PlantCARE (http://bioinformatics.psb.ugent.be/webtools/plantcare/html/ (accessed on 8 April 2022)) was then employed to locate the *cis*-elements in promoters of the *LkNF-YB* genes [59]. Promoter analysis maps of the *LkNF-YB* gene family were plotted with TBtools (version number 1.098769) by integrating abiotic stress and hormone-related elements.

### 4.5. Gene Ontology Annotation of LkNF-YBs

The gene function was annotated on eggNOG-mapper (http://eggnog-mapper.embl.de/ (accessed on 30 April 2022)), and the results were sorted and analyzed by TBtools to obtain the tables and bar graphs.

### 4.6. Protein Interaction Network Analysis

The amino acid sequence of the *LkNF-YB* proteins was used as a query to estimate the protein interaction network of *LkNF-YB* proteins using the STRING (version 11.0; https://string-db.org/cgi/input.pl (accessed on 5 May 2022)) database. The ‘organism’ option was set as ‘*Arabidopsis* thaliana’, and the basic settings included ‘evidence’ and ‘text-mining, experiments, databases, co-expression, neighborhood, gene fusion and co-occurrence’.

### 4.7. Stress Treatment

Japanese larch seeds collected at Qingshan National Larch Breeding Base were rinsed with running water for 5 days and sown in a sterilized 1:1 mixture of soil and sand. Pour the right amount of water, seal the planting box with plastic wrap, and prick the holes. The sowing box was cultured in a greenhouse with a temperature of 20 °C and a cycle of 16 h of light and 8 h of darkness. After the seeds germinate, remove the plastic wrap, keep the soil surface moist, increase the amount of watering after growing euphylla. Three-month-old larch seedlings with nearly 15 cm height were selected. Fifteen in vitro plants were divided into five groups. The seedlings were immersed in 1/2 MS liquid medium and then treated with 150 mM NaCl for salt stress or with 7 % (*w*/*v*) PEG6000 for drought stress. Other seedlings were subjected to hormone treatment; three kinds of hormones, including MeJA, ABA and SA, were used at 100 μM concentration. Then, each treatment lasted 24 h; their roots and leaves were collected at different time points of 3, 6, 12, 24 h. The collected samples were quickly frozen using liquid nitrogen and stored at −80 °C for further use. Untreated seedlings (0 h) were used as the control group, and 3 independent biological replicates were obtained for each group [60].

### 4.8. RNA Extraction and RT-qPCR Analysis

Total RNA was extracted from samples using the Batech Universal Plant Total RNA Rapid Extraction Kit (TRIplant RNA Isolation, Bio Teke, Beijing, China). Take 0.1 g of plant tissue per serving. The RNA concentration was determined, and the RNA quality was observed in 1% agarose gel. Single-stranded cDNA was synthesized using reverse transcription kit (MonScriptTM RTlll All-in-One Mix with dsDNase), and the synthesized cDNA was diluted 10 times for RT-qPCR template. ChamQ Universal SYBR qPCR Master Mix Kit (Vazyme, Nanjing, China) was used, and a 20 μ system RT-qPCR was performed according to the instructions. Six genes, including *LkNF-YB1*, *3*, *8*, *13*, *14* and *17,* were selected, and RT-qPCR specific primers were designed. α-Actin gene was selected as the internal reference gene. The reaction system consisted of 10 µL 2 × ChamQ Universal SYBR qPCR Master Mix, 0.4 µL (10 µM) forward primer, 0.4 µL (10 µM) reverse primer, 1 µL (100 ng) cDNA and 8.2 µL ddH_2_O (The primer sequence is shown in Appendix A of Appendix A). The reaction process is as follows: reaction at 95 °C for 30 s, 95 °C for 10 s, 60 °C for 30 s, 40 cycles. Each sample was set up for three biological replicates, and the expression levels of related genes were analyzed using 2^−ΔΔCT^. The significance of the differences between groups was evaluated using the *t*-test (* *p* < 0.05, ** *p* < 0.01).

## 5. Conclusions

To conclude, we identified 20 *LkNF-YB* family genes in *L. kaempferi* and conducted a series of bioinformatic analyses on their phylogenetic relationships, conserved motifs, promoter *cis*-acting elements and GO annotation. *LkNF-YB* proteins are putative non-LEC1 type NF-YB family transcription factors. Their promoters have various phytohormone- and stress-associated *cis*-acting elements. These proteins showed significant responses under various treatments with salt, drought, ABA, SA and MeJA. Among them, *LkNF-YB*3 revealed the strongest responses against drought and ABA treatments. The prediction of the protein interaction network for *LkNF-YB3* indicated that *LkNF-YB3* could interact with not only the stress-responsive proteins, such as heat shock proteins, but also the histone proteins to modulate epigenetic marks on the downstream signaling genes. Our work provides the basic knowledge on the *LkNF-YB* gene family, and these preliminary results would be the foundation for further in-depth functional characterization studies of the *LkNF-YB* gene family in *L. kaempferi*.

## Figures and Tables

**Figure 1 ijms-24-08910-f001:**
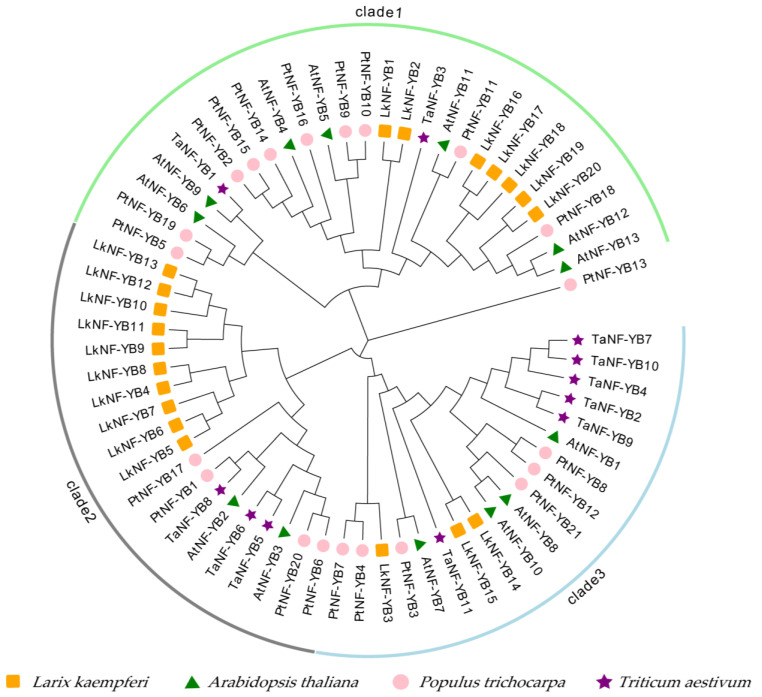
Phylogenetic analysis of NF-YB gene family members from *Larix kaempferi*, *Arabidopsis thaliana*, *Populus trichocarpa* and *Triticum aestivum*. MEGA7.0 was used to construct phylogenetic tree (the boot program value was 500 repeats), and the boot program value is shown on each branch.

**Figure 2 ijms-24-08910-f002:**
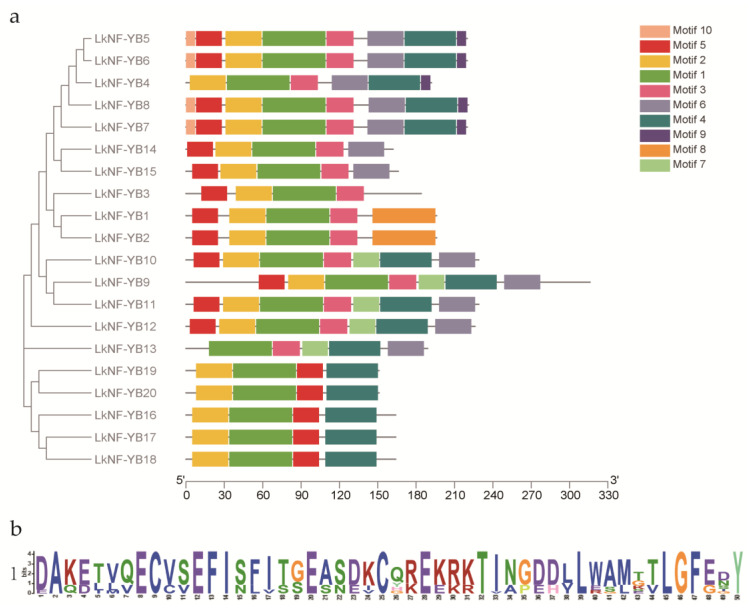
Phylogenetic relationships and conserved motifs of NF-YBs in *L. kaempferi*. (**a**) The conserved motifs were obtained from MEME website. Ten different kinds of conserved motifs were marked with different colors. (**b**) *LkNF-YB* conserved core consensus sequence. The overall height in each stack indicates sequence conservation at that position; the height of each residue letter indicates the relative frequency of the corresponding residue (color figure online).

**Figure 3 ijms-24-08910-f003:**
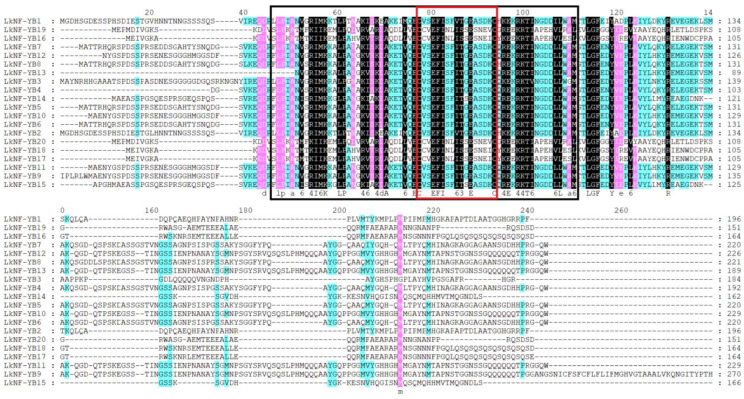
Multiple sequences alignments of NF-YBs in *L. kaempferi*. Different colors used to highlight amino acid residues represent degrees of amino acid identity. The black box represents the positions of conserved motifs; the red box shows the B domain.

**Figure 4 ijms-24-08910-f004:**
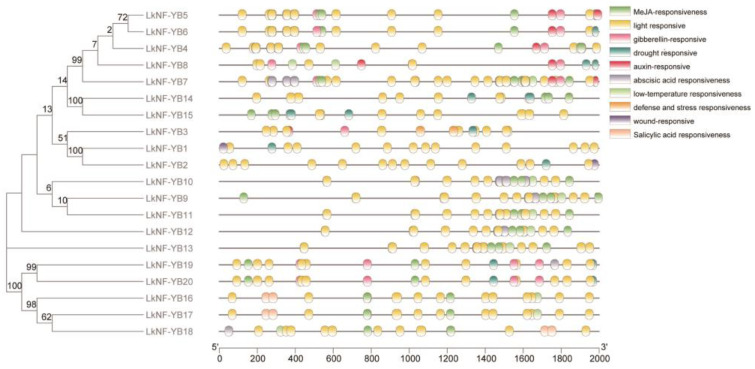
Abiotic- and phytohormone-related *cis*-elements in *LkNF-YB* gene promoters. Different colors represent different *cis*-elements.

**Figure 5 ijms-24-08910-f005:**
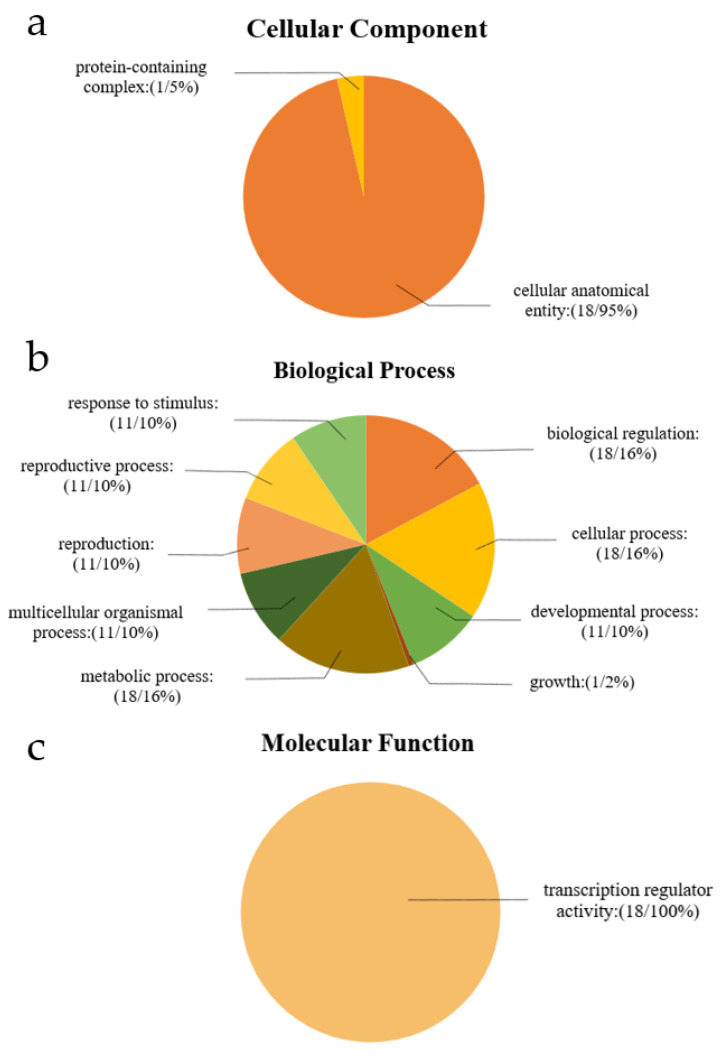
The gene ontology analysis of *LkNF-YB* gene family. The GO analysis of *LkNF-YB* genes predicted for their involvements in (**a**) cellular component, (**b**) biological processes and (**c**) molecular functions. (The number of genes enriched on the GO entry/The proportion of the GO entry to the GO category).

**Figure 6 ijms-24-08910-f006:**
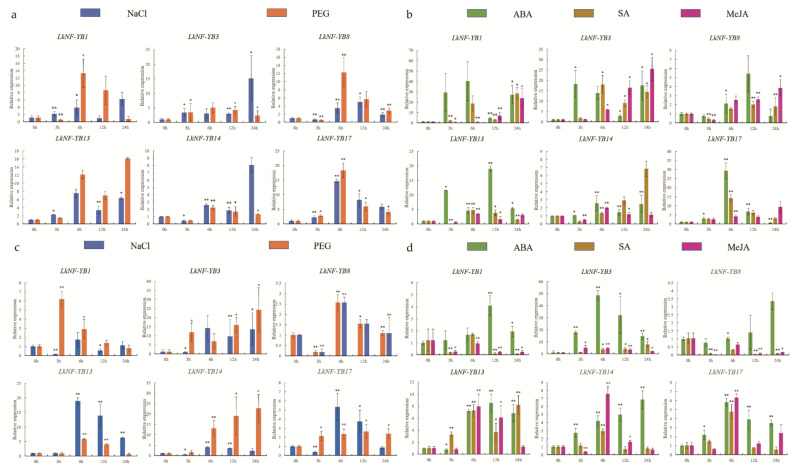
(**a**) The relative expression level of six *LkNF-YB* genes in leaves under salt and drought treatment by RT-qPCR. (**b**) The relative expression level of six *LkNF-YB* genes in leaves under ABA, SA and MeJA treatment by RT-qPCR. (**c**) The relative expression level of six *LkNF-YB* genes in roots under salt and drought treatment by RT-qPCR. (**d**) The relative expression level of six *LkNF-YB* genes in roots under ABA, SA and MeJA treatment by RT-qPCR. Blue and orange showed the expression pattern of *LkNF-YBs* under salt and drought treatment, respectively; Green, brown and pink showed the expression pattern of *LkNF-YBs* under ABA, SA and MeJA treatment, respectively. Error bars represent the deviations from three biological replicates. The x-axis represents the time points after treatment, and the asterisk indicates significant differences at *p* < 0.05 (*), *p* < 0.01 (**).

**Figure 7 ijms-24-08910-f007:**
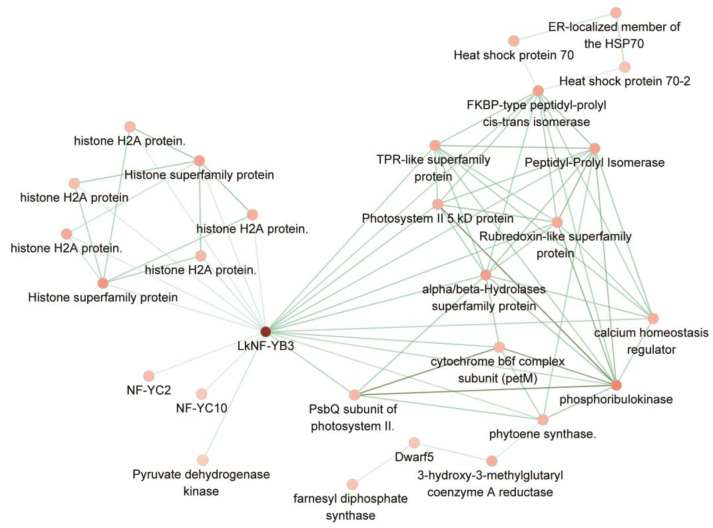
Protein interaction network of *LkNF-YB*3.

**Figure 8 ijms-24-08910-f008:**
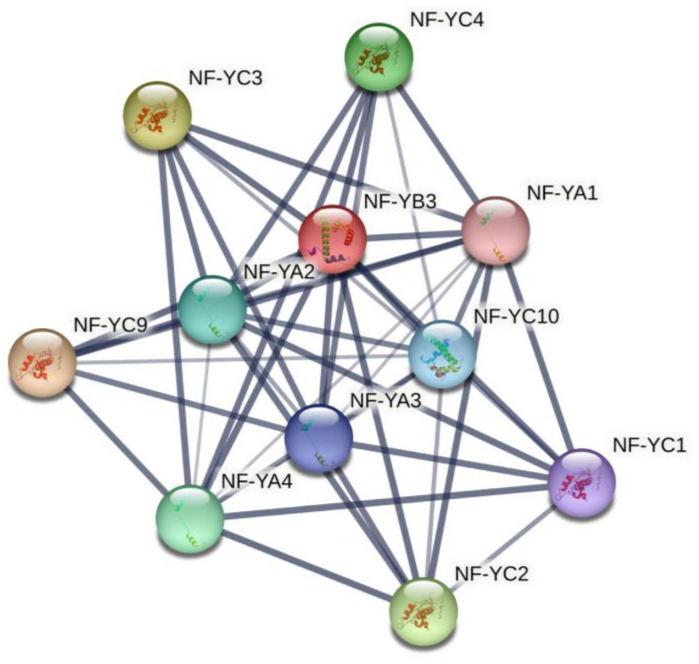
Protein interaction network of NF-YB. Different colored lines represent different evidence of an interaction.

**Table 1 ijms-24-08910-t001:** Parameters for the 20 identified NF-YB genes and deduced polypeptides present in the *L. kaempferi* genome.

Gene Name	Gene ID	Peptide (aa)	MW(Da)	PI	GRAVY	AI	II	Subcellular Location
*LkNF-YB1*	Gene10239	196	21,727.64	6.44	−0.696	65.26	59.28	nucleus
*LkNF-YB2*	Gene44058	196	21,769.72	6.45	−0.697	65.77	58.3	nucleus
*LkNF-YB3*	Gene38724	184	20,063.19	7.06	−0.903	57.34	48.31	nucleus, chloroplast, cytosol
*LkNF-YB4*	Gene36497	192	20,629.94	8.46	−0.722	55	41.54	nucleus
*LkNF-YB5*	Gene35188	220	23,731.13	7.76	−0.86	48.91	50.06	nucleus
*LkNF-YB6*	Gene30921	220	23,731.13	7.76	−0.86	48.91	50.06	nucleus
*LkNF-YB7*	Gene29425	220	23,713.1	7.76	−0.849	50.68	48.98	nucleus
*LkNF-YB8*	Gene32237	221	23,953.42	7.78	−0.858	50.9	51.1	nucleus
*LkNF-YB9*	Gene29143	316	34,583.76	8.93	−0.62	59.27	45.2	nucleus, extracellular
*LkNF-YB10*	Gene34173	229	24,879.45	6.08	−0.904	44.76	44.17	nucleus
*LkNF-YB11*	Gene27630	229	24,879.45	6.08	−0.904	44.76	44.17	nucleus
*LkNF-YB12*	Gene9593	226	24,548.07	6.34	−0.917	44.91	44.08	nucleus
*LkNF-YB13*	Gene13411	189	20,652.69	7.93	−0.886	47.51	36.88	nucleus
*LkNF-YB14*	Gene36418	162	18,033.17	5.83	−0.788	61.42	59.38	nucleus
*LkNF-YB15*	Gene41897	166	18,395.56	5.98	−0.789	60.54	59.35	nucleus
*LkNF-YB16*	Gene39876	164	18,739.08	5.51	−0.724	70.79	74.83	nucleus, extracellular, chloroplast, mitochondria
*LkNF-YB17*	Gene42279	164	18,779.19	5.52	−0.696	73.17	72.15	nucleus, extracellular, chloroplast, mitochondria
*LkNF-YB18*	Gene12177	164	18,739.08	5.51	−0.724	70.79	74.83	nucleus, extracellular, chloroplast, mitochondria
*LkNF-YB19*	Gene40159	151	16,977.12	4.69	−0.546	75.63	53.07	nucleus, chloroplast, cytosol, mitochondria
*LkNF-YB20*	Gene13131	151	16,977.12	4.69	−0.546	75.63	53.07	nucleus, chloroplast, cytosol, mitochondria

MW = Molecular weight; PI = Theoretical isoelectric; GRAVY = Grand average of hydropathicity; AI = Aliphatic index; II = Instability index.

## Data Availability

Not applicable.

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
