# Peer review of "Comprehensive Analysis of the NF-YB Gene Family and Expression under Abiotic Stress and Hormone Treatment in Larix kaempferi"

_ijms, 2023, doi:10.3390/ijms24108910_

Round 1
Reviewer 1 Report
- This paper correspond for scope of journal.
- The title corresponds to the content of the paper.
- This study represents a significant contribution to comprehensive knowledge of the L. kaempferi NF-YB gene family in the aim of improving L. kaempferi stress resistance by regulating NF-YB factors.
- The main question of paper addressed to study NF-YB, a subfamily of Nuclear Factor Y (NF-Y) transcription factors, plays crucial roles in many biological processes of plant growth and development and abiotic stress responses, which can therefore be good candidate factors for breeding stress-resistant plants..
- The aim of research is clearly and fully pointed out on the end of chapter of Introduction. Should be pointed out as particular paragraph at the end of Chapter of introduction
- Key words are appropriate, but should be write single key word.
- Scientific methodology is applied correctly for this type of study.
- Results are clearly presented and discussed.
- Tables, figures, pictures are clear.
- The conclusions are clear and based on research
- This study represents complementary to the previous ones.
- Manuscript is acceptable after minor corrections, but not in this Journal!
Author Response
Dear reviewer:
Thank you for your time and constructive comments on our manuscript.
Please see the attachment.

Reviewer 2 Report
Line 114: Table legend: The basic information...better to change...what mean basic?
Line 122: proteins are closely related to Arabidopsis proteins? As only two species are used..this interference is not correct. please rewrite.
line 124-125: motif 2 is also present in all..rewrite explaining motif distribution and tree grouping...
What is the difference between fig 2b and fig 3b? and why there is a difference with the same seq data?
Section 2.4: its general representation of data.. what is the difference among the family members? Provide a detail table in supplementary data.
Line 178-179: source of preliminary data? is this part of the current study? where is data?
Line 183: L. kaempferi make it italic
Need depth discussion on the role of genes referencing previous data on wheat, apple, Populus, and Arabidopsis...
Why only Arabidopsis is used in the phylogenetic tree. It would be useful to use all reported wheat, apple, Populus and compare the clustering with their expression in this and previous study.
cis-acting elements analysis is not utilized well. Authors can check promoters of the highly expressing or genes showing stress/organ-specific expression for binding sites and correlate it with their expression.
Methods
What does it mean ''13 amino acid sequences'' 13 aa from each genes? or total 13 gene's protein seq.? rewrite
Figure 6: what does WT indicate? I think better to use control/untreated. Long with gene name authors can add organ name eg LkNF-YB1_leaf.
Lines 398: primer details? can be added to the supplementary file.
Author Response

(The authors gave the same response as above.)

Reviewer 3 Report
The work “Comprehensive Analysis of the NF-YB Gene Family and Expression under Abiotic Stress and Hormone Treatment in Larix kaempferi” is interesting from the point of view of improving resistance in forest species. I have some comments to be considered by the authors:
Abstract: would be more attractive to mention the most relevant result from the gene expression pattern analysis.
Material and methods
Line 370: the link for eggnog-mapper does not work
Paragraph 4.7. The provenance, germination and growth conditions of the plants is omitted. Please, provide information about the experiment or cite the reference where it is described.
How many plants were subjected to each treatment?
Line 386: time points after treatment? the same time points for all the treatments?
Line 388: individual plants were considered as biological replicates?
Line 391: wat amount of tissue was used?
Paragraph 4.8. RAW data from gene expression analysis should be presented as supplementary material. Supplementary Materials link is not accessible.
Results
Line 106: review this sentence.
Figure 4. Correct alicylic acid please.
Paragraph 2.5. Correct typographical errors, such as “meta-bolic” throughout the paragraph.
Figure 5. Could you explain what the numbers in brackets mean, please?
Line 181: introduce “the” between “we investigated” and “effect”
Line 183: L. kaempferi in italics
Line 189: Is this sentence correct?
Line 200: remove the second LkNF-YB before 14
Discussion
Line 287-290: rewrite this sentence
Line 301-303: rewrite this sentence
Line 311: unexlored?
Author Response

(The authors gave the same response as above.)

Round 2
Reviewer 3 Report
The minority comments on the article have been addressed by the authors, however this has not been the case with the major concerns, such as those referring to the biological assay and access to raw data. If I would have access to the supplementary material, my doubts might be resolved.
I have still some comments to be considered by the authors:
Table 1. correct Locatioon
Line 120. Triticum
Line 166. Correct bio-logical
Figure 5 legend. The ratios between brackets must be explained.
Line 307. Remove investigated.
Paragraph “Stress treatment”. The provenance of the Larix kaempferi plant material, germination process and growth conditions is still missing.
The expression data of the 6 LkNF-YB genes analyzed under the different stress conditions (data corresponding to Figure 6) should be accessible as supplementary material. On the other hand, the supplementary material link provided in the Supplementary Material section still does not work.
Author Response
Dear reviewer,
Thank you for your time and constructive comments on our manuscript. We have made the modification.
Please see the attachment.
